# FROM DECOUPLING TO ADAPTIVE TRANSFORMATION: A WIDER OPTIMIZATION SPACE FOR PTQ

**Zhaojing Wen**[*], **Qiulin Zhang**[*], **Yuan Zhang, Rudan Chen, Xichao Yang, Di Xie, Jiang Zhu**[†]
Hikvision Research Institute, Hangzhou, China
`{wenzhaojing,zhangqiulin5,`
`zhangyuan,chenrudan,yangxichao,xiedi,zhujiang.hri}@hikvision.com`

## ABSTRACT

Post-Training low-bit Quantization (PTQ) is useful to accelerate DNNs due to its high efficiency, the current SOTAs of which mostly adopt feature reconstruction with self-distillation finetuning. However, when bitwidth goes to be extremely low, we find the current reconstruction optimization space is not optimal. Considering all possible parameters and the ignored fact that integer weight can be obtained early before actual inference, we thoroughly explore different optimization space by quant-step decoupling, where a wider PTQ optimization space, which consistently makes a better optimum, is found out. Based on these, we propose an Adaptive Quantization Transformation(AdaQTransform) for PTQ reconstruction, which makes the quantized output feature better fit the FP32 counterpart with adaptive per-channel transformation, thus achieves lower feature reconstruction error. In addition, it incurs negligible extra finetuning cost and no extra inference cost. Based on AdaQTransform, for the first time, we build a general quantization setting paradigm subsuming current PTQs, QATs and other potential forms. Experiments demonstrate AdaQTransform expands the optimization space for PTQ and helps current PTQs find a better optimum over CNNs, ViTs, LLMs and image super-resolution networks, e.g., it improves NWQ by 5.7% on ImageNet for W2A2-MobileNet-v2. Codes are available at https://github.com/zjxyz/AdaQTransform.

## 1 INTRODUCTION

Low-bit model quantization (quant) generally consists of Quantization-Aware Training (QAT) and Post-Training Quantization (PTQ). PTQ is almost the first choice for fast model quantization since it does not require the full training pipeline like labeled training data. Traditional PTQ (Krishnamoorthi, 2018) searches quant parameters through Mean Squared Error(MSE). which suffers from severe accuracy drop in 4 or 2 bits. With the help of gradient descent through self-distillation, current PTQs gradually narrow the accuracy gap, such as AdaRound (Nagel et al., 2020) for weight quantization, and BRECQ (Li et al., 2021) / NWQ (Wang et al., 2022a) for activation quantization. In quant params optimization, we can see they evolve closer and closer towards QAT, except tiny unlabeled calibration set and no need for original FP32 training pipeline. Their process is as shown in (a) and (b) of Figure.1: (i) freezes Conv's FP32 weight and bias; (ii) takes FP32 weight and FP32 activation as ground truth since PTQ owns no labeled dataset; (iii) iteratively searches an optimal weight's quant-step $s_w$ through the quant-error between FP32 weight and quantized weight, then freezes $s_w$ and (iv) PTQ reconstruction: optimize weight's AdaRound up-or-down parameter $\alpha$ and activation's quant-step $s_x$ per layer/block/network through quant-error between FP32 and quant output activation.

For weight quantization AdaRound, it optimizes up-or-down parameter $\alpha$ with weight's quant-step frozen to ensure a fixed integer base thus it can make a stable up or down rounding learning. However, differing from that integer activation has to be computed online during inference, integer weight is obtained early before inference and is fixed during inference, as described in Formula 4. With this ignored fact, still under a fixed integer weight base, there exists a wider optimization space for the de-quantized FP32 weight, making it not limited to an $\pm 1$ quant-step optimization distance, thus a

---

[*]Equal Contribution
[†]Corresponding Author

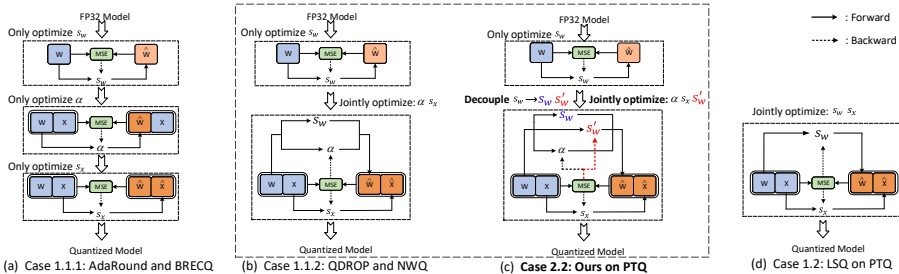

Figure 1: Different quantization settings with 'Case x.x.x' as detailed on Sec3.2. $W/X/s_w/\alpha/s_x$ denote weight/input/weight's quant-step/weight's adaround param/X's quant-step. Blue/orange box denote FP32/quantized values. The solid/dotted arrow line denote quantization forward/backward process. Note (c), we decouple the single $s_w$ of (b) into quant-step $s_w$ and de-quant-step $s_w'$.

lower reconstruction error might be met. That is, as shown in Formula 5, after computing the integer weight using $s_w$ during fake quantization simulation (quant-process), the $s_w$ used to convert integer weight into the FP32 counterpart (dequant-process) can be different. Or to say, the conventional weight's quant-step $s_w$ can be decoupled apart as quant-step $s_w$ and dequant-step $s_w'$ according to their different functions. The adaround learning process is still stable since the adaround learning base, integer weight, is still fixed. Although it is safe to decouple, can we obtain accuracy gain through decoupling? Further, now that PTQ evolves close to QAT, what will we gain if we directly optimize weight's quant-step into PTQ reconstruction like QAT as (d) of Figure.1? Considering all above, we thoroughly explore the setting of weight's quant-step into six possible cases through decoupling over various networks as Sec.3.2, where Case 1.1 is the current PTQ setting(AdaRound/BRECQ/NWQ), Case 1.2 is the current QAT's weight quant-step setting like LSQ (Esser et al., 2020) applied on PTQ reconstruction. We experimentally find a new setting Case 2.2, where we decouple quant-step, freeze integer weight and jointly optimize dequant-step as Formula 6, consistently performs the best.

At deeper optimization side, as Figure 2, Case 2.2 achieves a wider optimization space and makes quant output better fit the FP32 counterpart. The decoupled de-quant step $s_w'$ of weight finally achieves adaptive per-channel transformation on the output feature. Indeed, through visual and theoretical analysis, Case 2.2 equals to an adaptive per-channels linear transformation directly on output feature towards its FP32 counterpart. It is different from the common per-channel quant-step or offset like LSQ or LSQ+ (Bhalgat et al., 2020) applied on PTQ. Because they does not change the distribution of output feature, which only optimizes the current input while ignores current PTQ's output feature reconstruction. It is also different from SmoothQuant (Xiao et al., 2023), which transforms quant difficulty from activation to weight while also does not change the distribution of output. In addition, current PTQs freeze Conv's bias away from PTQ reconstruction. We find finetuning bias helpful, which provides further translating on quant output to narrow quant-error gap.

Considering above, we find the decoupling Case 2.2 and learnable bias $b$ can be derived into a more general form with introducing a scaling, $\epsilon$, and a translating factor, $\eta$, into current quantization settings, namely **AdaQTranform**, which is an easy integration to current PTQs. It incurs negligible extra finetuning cost and no extra inference cost. As Sec.3.3.2, AdaQTransform also subsumes normalization layer as its special form, but ours can be applied to networks/layers with or without normalization, e.g., low-level-vision-task nets like image super-resolution EDSR (Lim et al., 2017).

More importantly, AdaQTransform helps to build a general unified paradigm subsuming quantization settings of current PTQs, QATs, our new decoupling case and other cases. The detailed expansions from the unified paradigm to specific quant-settings can be seen in Sec.3.3. Our contributions are:

- We explore different optimization space by quant-step decoupling for PTQ. Based on the exploration, we propose AdaQTransform, which directly transforms the quant output closer to the FP32 counterpart(ground truth), making a wider optimization space and a narrower quant-error gap.

- Based on AdaQTransform, for the first time, we build a general unified paradigm subsuming quantization parameter settings for current PTQs, QATs and other possible cases.

- We evaluate AdaQTransform across CNNs, ViTs, LLMs and image super-resolution network EDSR, which proves AdaQTransform can be easily integrated to current PTQs, and consistently helps them to find a better optimum and better PTQ performance without extra inference cost.

## 2 RELATED WORK

Low bit model quantization is one of most effective technique in model compression (Zhang et al., 2020; Zhu et al., 2021; Han et al., 2020). Here we simply revisit its two main parts: PTQ and QAT.

**PTQ:** As one of the best weight PTQ, AdaRound (Nagel et al., 2020) proposed an adaptive rounding for weight rather than rounding-to-nearest operation. BRECQ (Li et al., 2021) found block-wise reconstruction behaves better than layer-wise ones. QDROP (Wei et al., 2022b) explored how activation quantization affected weight tuning, and proposed a random activation quantization during weight adaround learning. NWQ (Wang et al., 2022a) proposed a network-wise PTQ with inter-layer dependency. MRECG (Ma et al., 2023)/Bit-Shrink (Lin et al., 2023) tried to solve oscillation/sharpness problem. PD-Quant (Liu et al., 2023a) proposed to consider global information based on prediction difference metric. PTQ4ViT (Yuan et al., 2022), APQ-ViT (Ding et al., 2022), and RepQ-ViT (Li et al., 2023) tried to solve the PTQ for Vision Transformer(ViT).

Differently, we find weight's quant-step can be safely decoupled, and experimentally find the best case for PTQ reconstruction among six possible cases. Based on this best case, we theoretically propose an AdaQTransform to provide adaptive per-channel linear transformation on output feature.

**QAT:** Jacob et al. (2018b) proposed a fake quantizer simulation into QAT using gradient descent with straight-through estimator (STE). PACT (Choi et al., 2018) proposed parameterized clipping activation to learn the quantization range. LSQ (Esser et al., 2020) proposed to learn the quantization step directly, whose process applied on PTQ is shown as (d) of Figure.1. LSQ+(Bhalgat et al., 2020) further proposed a learnable offset. Nagel et al. (2022) tried to sovle oscillations in QAT. Wang et al. (2022b) learned lookup tables as quantizers. Liu et al. (2023b) proposed a data-free QAT for LLMs.

However, from Tab.1 Case 1.2, we find it is not suitable to directly borrow QAT's quant-step update setting into PTQ. This is because PTQ owns only a tiny unlabeled calibration set and no original FP32 training pipeline thus it need to freeze FP32 weight as ground truth, totally different from QAT.

## 3 METHOD

### 3.1 PRELIMINARIES AND CURRENT SOTA OF PTQ

As current PTQ SOTAs, we perform per-channel weight quantization and per-layer activation quantization. A classic linear symmetric PTQ process is as Formula (1,2,3). $s_w/s_x$, $w_l/w_u/x_l/x_u$ is the quant-step, upper/lower bound of quantization levels of FP32 weight $w$ and FP32 activation $x$. $\lfloor \cdot \rceil / \lfloor \cdot \rfloor$ indicates rounding/floor operation. $\lfloor \frac{w}{s_w} \rceil$ / $\hat{w}$ are called quantized(integer) / de-quantized weight. $h(\boldsymbol{\alpha})$ is AdaRound (Nagel et al., 2020) parameter of weight. $\sigma(\cdot)$ is Sigmoid function. They first initialize $s_w$ through minimizing the MSE between FP32 and quantized weight as Formula (1). Then freezing $s_w$ and optimizing AdaRound $h(\boldsymbol{\alpha})$ activation quantization as (2) through output feature reconstruction as (3). We can see FP32 weight $w$ and FP32 bias $b$ are freezed in current PTQs.

$$\hat{\boldsymbol{w}} = clip(\lfloor \frac{\boldsymbol{w}}{s_w} \rceil; w_l, w_u) \cdot s_w, \quad min_{s_w} ||\hat{\boldsymbol{w}} - \boldsymbol{w}||_F^2 \tag{1}$$

$$\hat{\boldsymbol{w}} = clip(\lfloor \frac{\boldsymbol{w}}{s_w} \rfloor + h(\boldsymbol{\alpha}); w_l, w_u) \cdot s_w; h(\alpha) = clip(\sigma(\alpha) * 1.2 - 0.1, 0, 1); \hat{\boldsymbol{x}} = clip(\lfloor \frac{\boldsymbol{x}}{s_x} \rceil; x_l, x_u) \cdot s_x \tag{2}$$

PTQ Reconstruction: $\hat{\boldsymbol{y}} = \hat{w} * \hat{x} + b = \sum((\lfloor \frac{\boldsymbol{w}}{s_w} \rfloor + h(\boldsymbol{\alpha})) \cdot s_w) * (\lfloor \frac{\boldsymbol{x}}{s_x} \rceil \cdot s_x); \quad \min_{\boldsymbol{\alpha}, s_x} ||\hat{\boldsymbol{y}} - \boldsymbol{y}||_F^2 \tag{3}$

**Fake and Real Quantization**. In order to better optimize PTQ parameters through gradient descent, the quantization function is simulated in FP32 format, denoted as the '*Fake Quant*' bracket of Formula 4. During practical inference acceleration, the FP32 simulation is converted to be integer-arithmetic-only (Jacob et al., 2018a) as the '*Real Quant*' bracket of Formula (4).

$$\boldsymbol{y} = \underbrace{\sum \boldsymbol{w} * x + b}_{FP32} \approx \underbrace{\sum \hat{\boldsymbol{w}} * \hat{\boldsymbol{x}} + b}_{Fake\ Quant, \text{for finetuning}} = \underbrace{s \cdot \sum \boldsymbol{w_{int}} * \lfloor \frac{\boldsymbol{x}}{s_x} \rceil + b}_{Real\ Quant, \text{for inference}} = \hat{y},$$

$$\text{where} \quad s = s_w \cdot s_x, \quad \boldsymbol{w_{int}} = clip(\lfloor \frac{\boldsymbol{w}}{s_w} \rfloor + \lfloor h(\boldsymbol{\alpha}) \rceil; w_l, w_u) \tag{4}$$

$\boldsymbol{w_{int}}$ is integer weight, which can be obtained early before practical fixed-point inference as the lower part of Formula (4).

## 3.2 Empirical Observations: Decouple Quant-Step of Weight

As the '$Real\ Quant$' of Formula 4, the $\boldsymbol{w_{int}}$ and $\boldsymbol{s_w}$ are determined before deployment, thus they can be treated as independent parameters. Given this property, for fake quantization, as Formula 5, we propose to decouple the quant-step of weight into quant-step $s_w$, which quantize FP32 weight to integer value, and dequant-step $s'_w$, which de-quantize integer weight back. Different from weight, quant-step of activation can not be decoupled since integer activation is different for different input.

$$\hat{\boldsymbol{w}} = clip(\lfloor \frac{\boldsymbol{w}}{s_w} \rfloor + h(\boldsymbol{\alpha}); w_l, w_u) \cdot s_w \quad \Rightarrow \quad \hat{\boldsymbol{w}} = clip(\lfloor \frac{w}{\boldsymbol{s_w}} \rfloor + h(\boldsymbol{\alpha}); w_l, w_u) \cdot s'_w \quad (5)$$

Under the condition where the quant-step of weight can be decoupled, for the first time, we fully explore different settings of weight's quant-step into six cases, based on whether quant-step is decoupled, and if decoupled, quant-step $s_w$ and de-quant step $s'_w$ are learnable or not after initialization.

- **Case 1**: the original single quant-step $s_w$ is not decoupled as convention.
  - ⊙ **Case 1.1**: not participates joint PTQ reconstruction optimization, as (a,b) of Fig. 1.
    - ◇ Case 1.1.1: Weight PTQ and activation PTQ are seperated, like AdaRound/BRECQ.
    - ◇ Case 1.1.2: Consider Weight AdaRound into activation PTQ, like QDROP/NWQ.
  - ⊙ **Case 1.2**: participates joint optimization during feature reconstruction, as (d) of Fig.1.
    - ◇ current QAT methods, like LSQ. **QAT's updating setting is not the best for PTQ**.
- **Case 2**: the original single quant-step $s_w$ is decoupled as two independent params $s_w$ and $s'_w$.
  - ⊙ **Case 2.1**: Only quant-step $s_w$ participates joint PTQ reconstruction optimization.
  - ⊙ **Case 2.2**: Only dequant-step $s'_w$ participates PTQ reconstruction optimization, as(c) of Fig.1.
    - ◆ frozen $s_w$: a fixed base for adaround learning;   learnable $s'_w$: adaptive transformation.
  - ⊙ **Case 2.3**: $s_w$ and $s'_w$ both participate in joint PTQ reconstruction optimization.

To evaluate their efficiency, we conduct experiments on MobileNet-v2, ResNet-18, and MnasNet2.0.

Table 1: Acc@1 on ImageNet among different quant-step settings across various nets.

| Methods | W/A | MobileNet-v2 | ResNet-18 | MnasNet2.0 |
|---|---|---|---|---|
| Case 1.1.1 (AdaRound, last PTQ SOTAs) | 3/2 | 0.32 | 41.65 | 1.07 |
| Case 1.1.2 (NWQ, current PTQ SOTAs) | 3/2 | 38.92 | 60.82 | 52.17 |
| Case 1.2 (LSQ, QAT's SOTA on PTQ) | 3/2 | 39.65 | 60.26 | 49.78 |
| Case 2.1 | 3/2 | 38.77 | 59.90 | 48.40 |
| **Case 2.2** | 3/2 | **42.60** | **61.06** | **54.19** |
| Case 2.3 | 3/2 | 41.42 | 60.86 | 49.33 |

We find out that Case 2.2, where we decouple the original quant-step $s_w$ as $s_w$ and $s'_w$ then make only dequant-step $s'_w$ learnablely participates joint PTQ reconstruction, shown as (c) of Figure 1, consistently provides the best performance, which even does a better job than Case 2.3. This is because (i) there is only a tiny unlabeled calibration set in PTQ, a frozen FP32 weight during finetuning makes the lowest quant-error. To further narrow quant-error gap, we need to apply AdaRound. and (ii) a stable AdaRound learning requires a fixed integer base but Case 2.1 and 2.3 bring fluid integer base.

Table 2: Visualization of Decoupling Case 2.2 during PTQ Reconstruction.

| | 0 | 5k | 15k | 20K | | 0 | 5k | 15k | 20K |
|---|---|---|---|---|---|---|---|---|---|
| $s_{w1}$ | 0.544 | 0.544 | 0.544 | 0.544 | $s_{w2}$ | 0.943 | 0.943 | 0.943 | 0.943 |
| $s'_{w1}$ | 0.544 | 0.508 | 0.444 | 0.442 | $s'_{w2}$ | 0.943 | 0.902 | 0.796 | 0.795 |
| Loss of Case 1.1.2 | 107 | 59.3 | 55.2 | 50.7 | Loss of Case 2.2 | 107 | **54.4** | **51.1** | **46.5** |

We visualize the learning process of the dequant-step $s'_w$ in Case 2.2 on W3A2 ResNet-18 as Tab. 2. At iteration 0, $s_w$ and $s'_w$ is decoupled from the the same value, then $s_w$ is frozen and $s'_w$ is learnable during PTQ reconstruction with weight's adaround param and activation quant-step. We can see Case 2.2's dequant-step $s'_w$ is updated accordingly and provides lower loss than current PTQ Case 1.1.2.

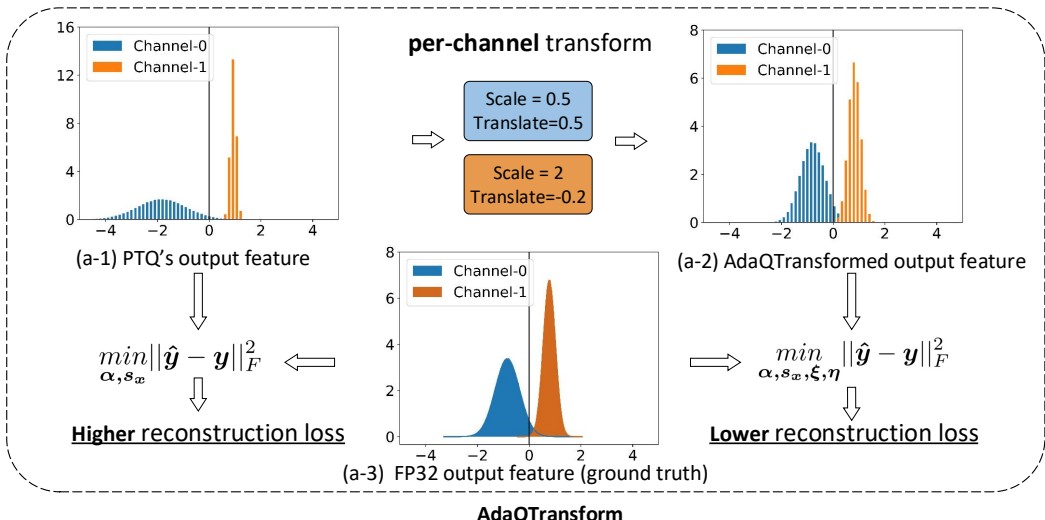

Figure 2: AdaQTransform: adaptive per-channel transformation on output feature.

## 3.3 FROM DECOUPLING TO ADAPTIVE QUANT TRANSFORMATION

Superficially, from empirical experiments in Sec.3.2, Case 2.2 consistently provides better accuracy and lower reconstruction loss. At deeper side, as Fig.2, Case 2.2 provides a better fit for each output channel to its FP32 ground truth. That is, Case 2.2 achieves adaptive per-channel scaling on the output feature. In theory, PTQ reconstruction in Formula.3 for Case 2.2 can be denoted as Formula.6, where $w_{floor} = \lfloor \frac{w}{s_w} \rfloor$ is frozen after initialization as Formula.1. Recall that We perform per-channel weight quantization and per-tensor activation quantization. Thus $s'_w$ is a vector with out-channels elements while $s_x$ is a scalar with one element. Therefore, the decoupled-out $s'_w$ theoretically provides direct per-channel scaling on output feature to further minimize the quant-error gap.

$$\hat{y} = s'_w \cdot \left\{ s_x \cdot \sum (w_{floor} + h(\alpha)) * \lfloor \frac{x}{s_x} \rceil \right\} + b, \quad \text{objective: } \min_{\alpha, s_x, s'_w} ||\hat{y} - y||_F^2 \quad (6)$$

A deeper look at Formula 6, there are two params, $w_{floor}$ and $b$, left frozen. $w_{floor}$ is frozen since a stable up or down rounding learning requires a fixed base. Bias $b$ is frozen due to inherited PTQ tradition. However, as we know, current PTQ works step closer and closer towards QAT. We find out it is enough to finetune bias $b$ with self-distillation between FP32 and quant output, which helps narrow the quantization error caused by quantized weight and quantized input, making their output fit closer to its FP32 counterpart, as shown in Fig.2. Therefore, in addition to per-channel scaling, our decoupling Case 2.2 can be further equipped with per-channel translating as

$$\hat{y} = s'_w \cdot \left\{ s_x \cdot \sum (w_{floor} + h(\alpha)) * \lfloor \frac{x}{s_x} \rceil \right\} + b', \quad \text{objective: } \min_{\alpha, s_x, s'_w, b'} ||\hat{y} - y||_F^2 \quad (7)$$

The PTQ reconstruction objective becomes the right of Formula 7. In order to be notation-consistent with previous methods as Formula 3, we introduce a per-channel scaling and translating factor, $\eta, \xi$, to denote Formula.7 as Formula.8, dubbed as **Ada**ptive **Q**uant **Transform**ation (**AdaQTransform**).

$$\hat{y} = \xi \cdot \left\{ \sum ((\lfloor \frac{w}{s_w} \rfloor + h(\alpha)) \cdot s_w) * (\lfloor \frac{x}{s_x} \rceil \cdot s_x) \right\} + b + \eta, \quad \text{objective: } \min_{\alpha, s_x, \xi, \eta} ||\hat{y} - y||_F^2 \quad (8)$$

### 3.3.1 NO EXTRA INFERENCE COST IN ADAQTRANSFORM

Although AdaQTransform add extra parameters into PTQ finetuning process, they will be merged into existing params thus cause no extra cost in actual inference. That is, when PTQ finetuning reconstruction process is finished, the finetuning Formula (8) can be inferred as Formula (9), which is as original inference Formula (4) like prior hardware-friendly PTQ works (Wang et al., 2022a).

$$\hat{y} = \widetilde{s} \cdot \left\{ \sum (w_{int} * \lfloor \frac{x}{s_x} \rceil) \right\} + \widetilde{b}, \quad \text{where } \widetilde{s} = \xi \cdot s_w \cdot s_x, \ \widetilde{b} = b + \eta \quad (9)$$

### 3.3.2 DIFFENRENCE BETWEEN ADAQTRANSFORM AND NORMALIZATION

Our AdaQTransform is significantly different from normalization, such as Batch/Group/Layer Normalizaion (BN/GN/LN), here we take BN as a example, in three points.

First, BN is used to stabilize FP32 model training while AdaQTransform is used to help quantized model better fit FP32 counterpart during self distillation. BN normalizes data with statistical mean and variance then accordingly scales and shifts them back. AdaQTransform directly learns a per-channel linear transformation from quantized feature to FP32 counterpart.

Second, current PTQ publications all fold BN into its proceeding Conv layer before quantization. Current QATs do not fold BN but they choose to update both statistical mean, variance and learnable scale, shift. As far as we know, there is no study exploring how BN influence PTQ up to now. For Conv-BN structures, if BN is not folded into its preceding Conv, and participates PTQ reconstruction, it achieves finer-grained per-channel transformation by making BN's per-channel scaling and translating parameter $\gamma$ and $\beta$ learnable as Formula (10) with frozen $s_w$ and frozen mean $\mu$ and var $\sigma^2$. Compare Formula (10) with Formula (8), it can be seen as a special form of our AdaQTransform.

$$\hat{\boldsymbol{y}} = \frac{\gamma}{\sqrt{\sigma^2 + \epsilon}} \cdot \left\{ \sum (\boldsymbol{w_{int}} \cdot \boldsymbol{s_w}) * (\lfloor \frac{\boldsymbol{x}}{s_x} \rceil \cdot \boldsymbol{s_x}) \right\} + \beta - \frac{\gamma\mu}{\sqrt{\sigma^2 + \epsilon}} + b, \text{objective: } \min_{\alpha, s_x, \gamma, \beta} ||\hat{\boldsymbol{y}} - \boldsymbol{y}||_F^2 \quad (10)$$

Thirdly, **AdaQTransform is both applicable to networks/layers with normalization or without normalization** like image super-resolution networks, e.g., EDSR (Lim et al., 2017), layers such as the latter Conv in Conv-BN-Conv, two Convs in Conv-(ReLU)-Conv, as demonstration in Sec.4

### 3.3.3 EASY INTEGRATION AND ENHANCEMENT TO CURRENT REDISTRIBUTION METHODS

The core process of current redistribution methods for quantization, like LSQ+ (Bhalgat et al., 2020) with translating offset $z_x$ or RepQ-ViT (Li et al., 2023) with the scaling vector $scale$, is as follows.

$$\text{LSQ+: } \hat{\boldsymbol{y}} = \hat{\boldsymbol{w}} * \hat{\boldsymbol{x}} + b = \sum (\lfloor \frac{\boldsymbol{w}}{\boldsymbol{s_w}} \rceil \cdot \boldsymbol{s_w}) * (\lfloor \frac{\boldsymbol{x} - z_x}{s_x} \rceil \cdot s_x + z_x) + b \quad (11)$$

$$\text{RepQ-ViT: } \hat{\boldsymbol{y}} = \hat{\boldsymbol{w}} * \hat{\boldsymbol{x}} + b = \sum (\lfloor \frac{\boldsymbol{w} \cdot \boldsymbol{scale}}{\boldsymbol{s_w}} \rceil \cdot \boldsymbol{s_w}) * (\lfloor \frac{\boldsymbol{x}/\boldsymbol{scale}}{s_x} \rceil \cdot s_x) + b \quad (12)$$

We see their redistribution on output will be recovered. Thus they do no change the distribution of output feature. Differently, our AdaQTransform is directly performed on output feature and changes the distribution of output features, which further narrows the quant-error gap between $y$ and $\hat{y}$, and is more suitable for PTQ's output feature reconstruction. Thus, it is an easy integration and enhancement to redistribution PTQs. Example for RepQ-ViT is as follows and experiments is as Tab.4

$$\hat{\boldsymbol{y}} = \xi \cdot \hat{\boldsymbol{w}} * \hat{\boldsymbol{x}} + b + \eta = \xi \cdot \sum ((\lfloor \frac{\boldsymbol{w} \cdot \boldsymbol{scale}}{\boldsymbol{s_w}} \rceil) \cdot \boldsymbol{s_w}) * (\lfloor \frac{\boldsymbol{x}/\boldsymbol{scale}}{s_x} \rceil \cdot s_x) + b + \eta \quad (13)$$

### 3.4 FROM ADAQTRANSFORM TO GENERAL QUANTIZATION PARADIGM

Formula 8 also builds a general quantization paradigm expressing quantization settings from current PTQs to current QATs and to our PTQ-suitable AdaQTransform. We can induct each one as follows,

- **From General Quant Paradigm to AdaRound/BRECQ**:
  $h(\alpha)$=AdaRound, $\xi = 1, \eta = 0$, objective: $\min_{s_w} ||\hat{\boldsymbol{w}} - \boldsymbol{w}||_F^2, \min_{\alpha} ||\hat{\boldsymbol{y}} - \boldsymbol{y}||_F^2$ then $\min_{s_x} ||\hat{\boldsymbol{y}} - \boldsymbol{y}||_F^2$.

- **From General Quant Paradigm to QDROP/NWQ**:
  $h(\alpha)$=AdaRound, $\xi = 1, \eta = 0$, objective: $\min_{s_w} ||\hat{\boldsymbol{w}} - \boldsymbol{w}||_F^2$ then $\min_{\alpha, s_x} ||\hat{\boldsymbol{y}} - \boldsymbol{y}||_F^2$

- **From General Quant Paradigm to LSQ on PTQ**:
  $h(\alpha) = Round(\frac{\boldsymbol{w}}{\boldsymbol{s_w}} - \lfloor \frac{\boldsymbol{w}}{\boldsymbol{s_w}} \rfloor), \xi = 1, \eta = 0$, objective: $\min_{s_w, s_x} ||\hat{\boldsymbol{y}} - \boldsymbol{y}||_F^2$.

- **From General Quant Paradigm to Our AdaQTransform**:
  $h(\alpha)$=AdaRound, $\xi, \eta$ learnable, objective: $\min_{s_w} ||\hat{\boldsymbol{w}} - \boldsymbol{w}||_F^2$ then $\min_{\alpha, s_x, \xi, \eta} ||\hat{\boldsymbol{y}} - \boldsymbol{y}||_F^2$.

- **From General Quant Paradigm to QAT's LSQ**:
  $h(\alpha) = Round(\frac{\boldsymbol{w}}{\boldsymbol{s_w}} - \lfloor \frac{\boldsymbol{w}}{\boldsymbol{s_w}} \rfloor), \xi = 1, \eta = 0$, objective: $\min_{w, s_x, s_w, b} CrossEntropy(\hat{Logit}_{last}, label)$.

---

**Algorithm 1:** AdaQTransform PTQ

---

**Input:** Pretrained FP32 Model $\{W^l\}_{l=1}^N$; calib set;
**Params :** Activation's quant-step $s_x$; $W$'s quant
param: $s_w, \alpha$. AdaQTranform: $\xi, \eta$

— — — — — — — — — — — — — — — — — — — — —

$1_{st}$: Iterative MSE Optimization for $s_w$ as (1).

— — — — — — — — — — — — — — — — — — — — —

$2_{nd}$: PTQ Reconstruction.
**for** $j = 1$ to $T$ iterations **do**
  **for** $i = 1$ to $N$ layers **do**
    # Get output from FP32 and quantized
    $\hat{W}^i \xleftarrow{s_w,\alpha} W^i$ as Formula.2
    $\hat{x}^i \xleftarrow{s_x} x^i$ as Formula.2
    $y^i = W^i * x^i + b^i$;
    $\hat{y}^i = \xi \cdot \hat{W}^i * \hat{x}^i + b^i + \eta$; # as Formula.8
    $\Delta_i = ||y^i - \hat{y}^i||_F^2$
  $\Delta = \sum \Delta_i$,    # Optimize $s_x, \alpha, \xi, \eta$ as (8)
**Output:** Quantized model

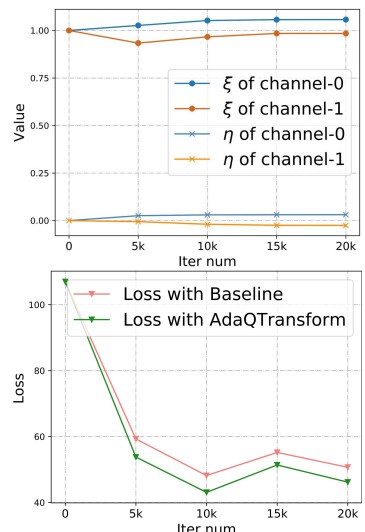

Figure 3: Visualization of AdaQTransform

## 4 EXPERIMENT

We evaluate AdaQTransform across various CNNs, ViTs, LLMs and image super-resolution networks using Pytorch (Paszke et al., 2019). Experimental settings are kept the same as each baselines. By convention, the first and last layer are quantized into 8 bits. AdaQTransform as Formula.8 is adopted. Integer inference with acceleration is performed with TVM on practical hardware.

### 4.1 EXPERIMENTS ON IMAGENET AND MS COCO

Table 3: Acc@1 on ImageNet among current PTQ methods.

| Methods | W/A | Mobile-v2 | Res-18 | Reg-600 | Mnas2.0 |
|---|---|---|---|---|---|
| Full Prec. | 32/32 | 72.49 | 71.08 | 73.71 | 76.68 |
| AdaRound(Nagel et al., 2020) | 4/4 | 64.33 | 69.36 | - | - |
| AdaQuant(Hubara et al., 2021) | 4/4 | 47.16 | 69.60 | - | - |
| BRECQ(Li et al., 2021) | 4/4 | 66.57 | 69.60 | 68.33 | 73.56 |
| QDROP(Wei et al., 2022b) | 4/4 | 68.84 | 69.62 | 71.18 | 73.71 |
| PD-Quant (Liu et al., 2023a) | 4/4 | 68.33 | 69.30 | 71.04 | 73.30 |
| MRECG (Ma et al., 2023) | 4/4 | 68.84 | 69.46 | 71.22 | - |
| NWQ (Wang et al., 2022a) | 4/4 | 69.14 | 69.85 | 71.92 | 74.60 |
| **AdaQTransform$_{\text{NWQ}}$(ours)** | **4/4** | **70.01** | **69.88** | **71.97** | **74.80** |
| BRECQ(Li et al., 2021) | 3/3 | 23.41 | 65.87 | 55.16 | 49.78 |
| QDROP(Wei et al., 2022b) | 3/3 | 57.98 | 66.75 | 65.54 | 66.81 |
| PD-Quant (Liu et al., 2023a) | 3/3 | 57.64 | 66.12 | 65.09 | 64.88 |
| MRECG (Ma et al., 2023) | 3/3 | 58.40 | 66.30 | 66.08 | - |
| NWQ (Wang et al., 2022a) | 3/3 | 61.24 | 67.58 | 67.38 | 68.85 |
| **AdaQTransform$_{\text{NWQ}}$(ours)** | **3/3** | **63.44** | **67.73** | **67.81** | **69.52** |
| BRECQ(Li et al., 2021) | 2/2 | 0.24 | 42.54 | 3.58 | 0.61 |
| QDROP(Wei et al., 2022b) | 2/2 | 13.05 | 54.72 | 41.47 | 28.77 |
| PD-Quant (Liu et al., 2023a) | 2/2 | 13.67 | 53.14 | 40.92 | 28.03 |
| MRECG (Ma et al., 2023) | 2/2 | 14.44 | 54.46 | 43.67 | - |
| NWQ (Wang et al., 2022a) | 2/2 | 26.42 | 59.14 | 48.49 | 41.17 |
| **AdaQTransform$_{\text{NWQ}}$(ours)** | **2/2** | **32.19** | **60.12** | **51.20** | **44.54** |

We first experiment on ImageNet classification task over various CNNs and vision transformers as shown in Tab.(3,4). The calibration set consists of 1024 unlabeled images randomly selected from the training set. We adopt Adam optimizer, the same learning rate as (Wei et al., 2022b; Ma et al., 2023) and 20k iterations for network-wise PTQ reconstruction as (Wang et al., 2022a). The average experimental results over 5 runs are summarized in Tab.3. In W4A4, our method provides about 0-1% Acc@1 improvement compared to the strong baseline including NWQ (Wang et al., 2022a), MRECG (Ma et al., 2023). In W3A3, our method improve MobileNet-v2 by 2.2% and MnasNet2.0 by 0.67%. In W2A2, where BRECQ shows nearly 0% Acc@1 on Mobile-v2 and Mnas2.0, our method still far outperforms NWQ by more than 3% on Mobile-v2, and Mnas2.0.

Table 4: Acc@1 on ImageNet for ViTs and DeiTs.

| Methods | W/A | ViT-S | ViT-B | DeiT-S | DeiT-B |
|---|---|---|---|---|---|
| FP32 | 32/32 | 81.39 | 84.54 | 79.80 | 81.80 |
| RepQ-ViT (Li et al., 2023) | 6/6 | 80.43 | 83.62 | 78.90 | 81.27 |
| **AdaQTranform$_{RepQ-ViT}$** | 6/6 | **80.59** | **83.89** | **79.12** | **81.53** |
| PTQ4ViT (Yuan et al., 2022) | 4/4 | 42.57 | 30.69 | 34.08 | 64.39 |
| APQ-ViT (Ding et al., 2022) | 4/4 | 47.95 | 41.41 | 43.55 | 67.48 |
| NWQ (Wang et al., 2022a) | 4/4 | 57.79 | 56.87 | 65.76 | 76.06 |
| **AdaQTransform$_{NWQ}$** | 4/4 | **58.12** | **57.24** | **66.34** | **76.20** |
| RepQ-ViT (Li et al., 2023) | 4/4 | 65.05 | 68.48 | 69.03 | 75.61 |
| **AdaQTranform$_{RepQ-ViT}$** | 4/4 | **70.40** | **76.47** | **73.50** | **78.93** |

For vision transformers, we experiments on ViT (Dosovitskiy et al., 2021) and DeiT (Touvron et al., 2021) as Tab.4. Our AdaQTransform outperforms NWQ by 0.5%, and outperforms PTQ4ViT (Yuan et al., 2022) and APQ-ViT (Ding et al., 2022) by a large margin, about 10%-32% better. Then we apply AdaQTranform into RepQ-ViT. We can see AdaQTransform helps RepQ-ViT improve about 4% in W4A4 and 0.3% in W6A6. Thus it demonstrates AdaQTranform helps narrow the quant-error gap between the quantized and FP32 activation.

Table 5: mAP on MS COCO for object detection.

| Methods | W/A | Faster RCNN | | RetinaNet | |
|---|---|---|---|---|---|
| | | ResNet-50 | ResNet-18 | ResNet-50 | MobileNet-v2 |
| FP32 | 32/32 | 40.26 | 34.91 | 37.39 | 33.31 |
| BRECQ (Li et al., 2021) | 4/4 | 37.19 | 33.41 | 34.67 | 29.81 |
| QDROP (Wei et al., 2022b) | 4/4 | 38.53 | 33.57 | 35.81 | 31.47 |
| NWQ (Wang et al., 2022a) | 4/4 | 38.54 | 33.63 | 35.98 | 31.81 |
| **AdaQTransform$_{NWQ}$(ours)** | 4/4 | **38.62** | **33.87** | **35.96** | **31.93** |
| QDROP (Wei et al., 2022b) | 3/3 | 33.49 | 31.21 | 32.13 | 27.55 |
| NWQ (Wang et al., 2022a) | 3/3 | 35.25 | 31.88 | 32.45 | 28.43 |
| **AdaQTransform$_{NWQ}$(ours)** | 3/3 | **35.72** | **32.25** | **32.48** | **28.86** |
| QDROP (Wei et al., 2022b) | 2/2 | 21.05 | 21.95 | 20.27 | 12.01 |
| NWQ (Wang et al., 2022a) | 2/2 | 25.01 | 23.92 | 22.95 | 16.21 |
| **AdaQTransform$_{NWQ}$(ours)** | 2/2 | **27.79** | **26.10** | **24.13** | **18.10** |

For object detection, we experiments on one-stage RetinaNet (Lin et al., 2017) and two-stage Faster RCNN (Ren et al., 2015), where Res-18, Res-50 and Mobile-v2 are selected as backbones respectively. As (Wei et al., 2022b; Li et al., 2021), we quantize the input and output layers of the network to 8 bits, do not quantize the head of the detection model, and quantize the neck (FPN). Results are shown in Tab.5. In W3A3 setting, AdaQTransform improves the mAP of Res-50-based Faster RCNN by 0.5% and Mobile-v2-based RetinaNet by 0.4%. In harder W2A2 setting, AdaQTransform achieves about 1% mAP improvement over the current best method across all four experimental networks, which obtains a 2.78% improvement on Res-50-based Faster RCNN.

## 4.2 EXPERIMENTS ON IMAGE SUPER-RESOLUTION. (NETWORKS WITHOUT NORMALIZATION)

As shown on Sec. 3.3.2, layer normalization can be seen as a special form of our AdaQTransform. To show the effectiveness of AdaQTransform on networks without layer normalization, we experiment on image super-resolution networks, i.e., EDSR of scale 4 (Lim et al., 2017). We borrow base code from AdaBM (Hong & Lee, 2024) and follow all the same settings except that we apply our AdaQTransform to AdaBM. The calibration dataset, 100 LR images, is randomly sampled from the DIV2K (Timofte et al., 2017) training dataset. The quantization range for activation is initialized using MinMax and quantization step for weight is initialized by OMSE. Then we freeze the network weights and finetune the quantization parameters for 10 epochs using Adam optimizer. For evaluation metrics, we measure reconstruction accuracy using the peak signal-to-noise ratio (PSNR) and the structural similarity index (SSIM) on Set5/Set14/Urban100/BSD100 (Huang et al., 2015). To compare the computational complexity of the quantized network, we report the feature average bit-width (FAB) that is averaged throughout the images of the test dataset.

As shown in Tab. 6, where AdaQTrans[†] denotes we apply our AdaQTransform to AdaBM (Hong & Lee, 2024), we can see our AdaQTransform consistently helps AdaBM improve the PSNR and SSIM on 4 experimental test sets and 4/3/2-bit quantization settings. Tab. 6 demonstrates our AdaQTransform differs from layer normalization and gains from a wider optimization space: it helps the quant output feature better fit the FP32 counterpart and achieves lower PTQ quantization error.

Table 6: PTQ for EDSR of scale 4.

| Model | W/A | Set5 | | Set14 | | BSD100 | | Urban100 | |
|---|---|---|---|---|---|---|---|---|---|
| | | FAB | PSNR/SSIM | FAB | PSNR/SSIM | FAB | PSNR/SSIM | FAB | PSNR/SSIM |
| EDSR (×4) | 32/32 | 32 | 32.10 / 0.893 | 32 | 28.57 / 0.781 | 32 | 27.56 / 0.736 | 32 | 26.02/ 0.784 |
| AdaBM-paper | 4/4MP | 3.8 | 31.02 / 0.860 | 3.7 | 27.87 / 0.751 | 3.5 | 26.91 / 0.700 | 3.7 | 25.11 / 0.736 |
| AdaQTrans[†] | 4/4MP | **3.7** | **31.17 / 0.870** | **3.5** | **27.99 / 0.761** | **3.5** | **27.04 / 0.713** | **3.7** | **25.03 /0.742** |
| AdaBM | 4/4 | 4 | 29.42 / 0.821 | 4 | 26.81 / 0.724 | 4 | 26.44 / 0.687 | 4 | 23.87 / 0.685 |
| AdaQTrans[†] | 4/4 | **4** | **31.41 / 0.845** | **4** | **27.55 / 0.742** | **4** | **26.81 / 0.698** | **4** | **24.64 / 0.718** |
| AdaBM | 3/3 | 3 | 28.93 / 0.804 | 3 | 26.49 / 0.711 | 3 | 26.24 / 0.679 | 3 | 23.53 / 0.667 |
| AdaQTrans[†] | 3/3 | **3** | **29.01 / 0.810** | **3** | **26.55 / 0.716** | **3** | **26.27 / 0.683** | **3** | **23.56 / 0.672** |
| AdaBM | 2/2 | 2 | 28.76 / 0.791 | 2 | 26.38 / 0.702 | 2 | 26.17 / 0.673 | 2 | 23.45 / 0.657 |
| AdaQTrans[†] | 2/2 | **2** | **28.84 / 0.802** | **2** | **26.44 / 0.712** | **2** | **26.23 / 0.681** | **2** | **23.46 / 0.666** |

As shown in Fig.4, our AdaQTransform helps AdaBM produces visually better reconstructed images with more details, e.g., AdaBM is relatively slur on the arch curve while AdaQTransform is clearer.

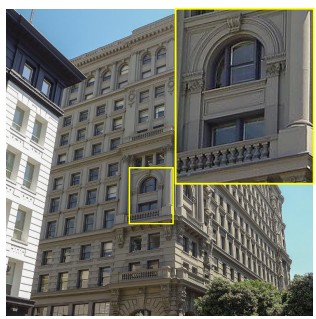 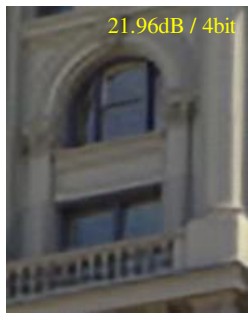 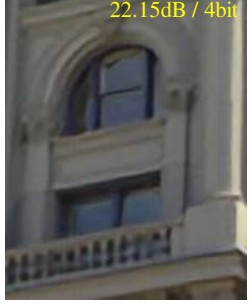

GT(img014)  AdaBM  AdaQTransform

Figure 4: Qualitative results on Urban100 with 4-bit EDSR-based models.

Table 7:AdaQTransform on LLMs LAMBADA.

| Method | W/A | Opt-1.3B | Opt-6.7B |
|---|---|---|---|
| FP32 | 32/32 | 72.0% | 79.8% |
| Naive | 8/8 | 69.1% | 41.9% |
| SmoothQuant | 8/8 | 70.8% | 80.0% |
| AdaQTransform[†] | 8/8 | **71.2%** | **80.1%** |
| SmoothQuant | 6/6 | 66.2% | 75.4% |
| OS+ | 6/6 | 66.4% | 75.5% |
| AdaQTransform[†] | 6/6 | **67.5%** | **75.6%** |

Table 8: Exploration for BN and AdaQTransform

| Methods | W/A | Mobile-v2 | Res-18 | Reg-600 |
|---|---|---|---|---|
| NWQ(BN-Floded) | 3/3 | 61.24 | 67.58 | 67.38 |
| BN-Not-Folded | 3/3 | 63.26 | 67.67 | 67.65 |
| Decoupling | 3/3 | 63.17 | 67.64 | 67.42 |
| AdaQTransform | 3/3 | **63.44** | **67.73** | **67.81** |
| NWQ(BN-Floded) | 2/2 | 26.42 | 59.14 | 48.49 |
| BN-Not-Folded | 2/2 | 32.09 | 60.09 | 51.18 |
| Decoupling | 2/2 | 31.43 | 59.91 | 50.32 |
| AdaQTransform | 2/2 | **32.19** | **60.12** | **51.20** |

## 4.3 EXPERIMENTS ON LARGE LANGUAGE MODELS (LLM)

As Tab.7, we compare AdaQTransform with SmoothQuant (Xiao et al., 2023) and OS+ (Wei et al., 2023) on LLM models Opt-1.3B/6.7B, by simple evaluation metric "Last Token Prediction Accuracy" on LAMBADA dataset's validation set. AdaQTransform[†] denotes AdaQTransform applied on SmoothQuant. We can see AdaQTransform brings SmoothQuant 1.3% gain on W6A6 Opt-1.3B.

## 4.4 ABLATION STUDY ON IMAGENET

4.4.1) **AdaQTransform V.s. BN**: For networks with normalization(BN/GN/LN), as we know, there have not been an academic PTQ work exploring BN's folding or not. Thus we explore it as Tab.8 based on NWQ which adopts BN-Folded setting by default. For BN-Not-Folded, we jointly optimize BN's learnable params $\gamma$ and $\beta$. It provides almost the same accuracy whether to update BN's statistical params $\mu, \sigma$ or not. We see BN-Not-Folded provides better performance than BN-Folded (NWQ). AdaQTransform provides tiny better accuracy than BN-Not-Folded, since AdaQTransform can be applied on layers with BN (equals to BN-Not-Folded) and other layers without BN. Therefore, as Sec.3.3.2, **AdaQTransform subsumes BN-Not-Folded, and covers a wider application range**.

4.4.2) **Visualization for AdaQTransform**: As Fig.3.4, AdaQTransform achieves adaptive per-channel transformation with adaptive $\xi, \eta$ for output channels, and converges to a lower loss.

## 4.5 TRAINING AND INFERENCE COST COMPARISON

For training, we only integrate AdaQTransform into each PTQ baselines and keep other things unchanged. The training time and memory is almost the same as the baseline. For inference, We perform pure 4/8-bit integer inference with TVM on hardware with code from HAWQ (Yao et al., 2021). The first and last layer are in 8 bits. The middle layer convolution is 4bits. The average inference time per batch over 30 measurement, each with 50 inferences, is as follows. We can see AdaQTranform improves accuracy without extra inference cost, and almost without training cost.

Table 9: W4A4 MobileNet-v2 training and inference cost

| Net | Acc@1 | Train-Time | Infer-Time (Batch=8) | Params | Params*Bit | FLOPs*Bit |
|---|---|---|---|---|---|---|
| NWQ | 69.14% | 46.5 min | 1.28 ms | 3.51 M | 20.79 M | 6.36 G |
| AdaQTransform$_{NWQ}$ | **70.01%** | 46.9 min | 1.28 ms | 3.51 M | 20.79 M | 6.36 G |

## 5 CONCLUSION

In this paper, we propose a novel PTQ approach, AdaQTransform, based on full exploration on weight's quantization step through decoupling. It can be easily integrated into current PTQ methods, expands their optimization space and helps the quantized output better fit the FP32 output, thus achieves lower PTQ feature reconstruction error. It incurs negligible extra finetuning cost and no extra inference cost. For the first time, AdaQTransform builds a general paradigm in quantization parameter update settings from current PTQs to QATs. Experiments on CNNs, ViTs, LLMs, and image super-resolution EDSR demonstrate AdaQTransform sets up a new PTQ SOTA.

ACKNOWLEDGMENTS

This work was supported by the National Key Research and Development Program of China under Grant 2020AAA0109004.

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

## A ADDITIONAL TRAINING OVERHEAD FOR ADAQTRANSFORM

Our AdaQTransform can be easily integrated into existing PTQ methods. It takes the same training iteration(20K), the same optimizer(Adam), and learning rate as their baseline. As Table 10, the training time and memory cost for extra introduced $\xi$ and $\eta$ can be ignored compared to their baseline.

Table 10: W4A4 Training Time and Memory for AdaQTransform

| Method | ResNet-18 Train-Time | ResNet-18 Train-Mem | MobileNet-v2 Train-Time | MobileNet-v2 Train-Mem |
|---|---|---|---|---|
| BRECQ (Li et al., 2021) | 18.4 min | 2665 M | 43.9 min | 5794 M |
| AdaQTransform$_{BRECQ}$ | 18.7 min | 2667 M | 44.8 min | 5797 M |
| NWQ (Wang et al., 2022a) | 46.5 min | 3267 M | 66.6 min | 7529 M |
| AdaQTransform$_{NWQ}$ | 46.9 min | 3272 M | 67.3 min | 7537 M |
| PD-Quant (Liu et al., 2023a) | 97.9 min | 3589 M | 184.0 min | 6759 M |
| AdaQTransform$_{PD\_Quant}$ | 98.5 min | 3592 M | 186.6 min | 6764 M |

## B QUANT-PARAM SETTING VS TRAINING PIPELINE FOR PTQ AND QAT

Quantization consists of two parts: training pipeline and quantization parameter update setting(quant-param setting). Their main differences over PTQ and QAT are as follows,

- Training Pipeline:
  - ⊙ Dataset:
    - ◇ PTQ: tiny unlabeled calibration set, e.g., 1024 unlabeled images randomly selected from ImageNet training set.
    - ◇ QAT: full labeled training set, e.g., 1280000 labeled images
  - ⊙ Training method:
    - ◇ PTQ: self-distillation to FP32 models, usually called finetuning. FP32 weight is usually frozen, or minor adjusted.
    - ◇ QAT: training with loss to labeled ground truth, FP32 weight is learnable with no limit.
  - ⊙ Training time:
    - ◇ PTQ: usually in minutes level.
    - ◇ QAT: usually in days level.
- Quant-param setting:
  - ⊙ PTQ: AdaRound (Nagel et al., 2020), BRECQ (Li et al., 2021), QDROP (Wei et al., 2022a), NWQ (Wang et al., 2022a), AdaBM (Hong & Lee, 2024)
  - ⊙ QAT: LSQ (Esser et al., 2020), LSQ+ (Bhalgat et al., 2020), CABM (Tian et al., 2023)

Here we experiment AdaQTransform's quant-param setting on QAT's training pipeline, and classic QAT's quant-param setting, LSQ, on PTQ's training pipeline as Table 11 for W2A2 MobileNet-v2 on ImageNet. We can see different quant-param setting suits different training pipeline.

Table 11: W2A2 MobileNet-v2: Quant-Param Setting vs Training Pipeline For PTQ and QAT

| Acc@1/Training-Time | PTQ training pipeline | QAT training pipeline |
|---|---|---|
| LSQ's quant-param setting | 27.65% / 66 min | **47.96% / 98 hours** |
| AdaQTransform's quant-param setting | **32.19% / 67 min** | 47.72% / 98 hours |

## C  ADAQTRANSFORM ON WEIGHT-ONLY PTQ

We conduct weight-only quantization experiments on CNNs, ViTs as Table 12.

Table 12: AdaQTransform on Weight-Only PTQ over ViTs

| Methods | W/A | ViT-S | ViT-B | DeiT-S | DeiT-B |
|---|---|---|---|---|---|
| FP32 | 32/32 | 81.39% | 84.54% | 79.80% | 81.80% |
| RepQ-ViT (Li et al., 2023) | 4/32 | 75.31% | 78.34% | 75.10% | 78.48% |
| **AdaQTransform$_{\text{RepQ-ViT}}$** | **4/32** | **79.39%** | **82.57%** | **78.12%** | **81.10%** |
| RepQ-ViT (Li et al., 2023) | 6/32 | 81.01% | 84.33% | 79.50% | 81.67% |
| **AdaQTransform$_{\text{RepQ-ViT}}$** | **6/32** | **81.16%** | **84.39%** | **79.78%** | **81.78%** |

## D  ADAQTRANSFORM GRANULARITY

Except for per-channel AdaQTransfom, we also explored other granularity, e.g., per-layer, per-pixel and per-network. Given output feature shape $C_{out} \times H \times W$,

- Per-channel $\xi, \eta$, what our AdaQTransform adopts, are two vectors, each with $C_{out}$ independent values for output feature of each Conv/FC layer.
- Per-layer $\xi, \eta$ are two scalar, with only 2*1 independent value, for the whole output feature of each Conv/FC layer.
- Per-pixel $\xi, \eta$ set each scalar for each pixel, with a matrix with $C_{out} \times H \times W$ independent values for each Conv/FC layer.
- Per-network is only one $\xi, \eta$ for the whole network, here we put them on the final FC layer, as two vectors with shape $C_{out}$

Experiments on W2A2 MobileNet-v2 is as Table 13, where per-channel is the best among all settings which incur no extra inference cost.

Table 13: AdaQTransform Granularity, Per-channel/layer/pixel/network, on MobileNet-v2

| Method | W/A | Acc@1 for MobileNet-v2 | No extra cost in inference |
|---|---|---|---|
| NWQ-baseline | 2/2 | 26.42 | ✔ |
| **per-channel (ours)** | 2/2 | **32.19** | ✔ |
| per-layer | 2/2 | 26.92 | ✔ |
| per-pixel | 2/2 | 32.36 | ✘ |
| per-network | 2/2 | 26.44 | ✔ |

