# OpenReview forum: "From Decoupling to Adaptive Transformation: a Wider Optimization Space for PTQ"
_ICLR.cc/2025/Conference — ICLR 2025 Poster_

### Official Review · Reviewer_8Mvu · 2024-10-27

**Soundness:** 4
**Presentation:** 3
**Contribution:** 3
**Rating:** 6
**Confidence:** 4

**Summary:**

This paper studies the problem of Post-training low-bit quantization (PTQ) and makes an interesting observation that wider optimization space can lead to better PTQ solutions. The key idea is to unify PTQ with Quantization-Aware Training (QAT) by introducing a scaling and a translating factor. The developed Adaptive Quantization Transformation(AdaQTransform) has been tested on several network architectures including CNNs, ViTs, and LLMs. The reported experimental results are unanimously promising and convincing.

**Strengths:**

+ Originality: A notable contribution is the unified framework subsuming PTQs and QATs in the literature. Such a unified treatment is beneficial for improving our understanding of the existing works on network compression scattered in the literature.
+ Quality: the overall quality of this paper is good. In particular, I appreciate the authors' empirical observations in Sec. 3.2, which helps the explanation of decoupling and the motivation behind AdaQTransform.
+ Clarity: the paper is relatively easy to follow in most parts. The reported experimental results are comprehensive and convincing. Most figures and tables are self-contained.

**Weaknesses:**

- Significance: "AdaQTransform improves the current best PTQ by 5.7% on W2A2-MobileNet-v2." This claim is vague because it does not mention 5.7% corresponds to which aspect of the performance. I fail to identify the corresponding discussion in Sec. 4. Accordingly, it is difficult to assess the significance of this work.
- Experimental results. The benchmark methods used on MS COSO for object detection in Table 3 are all published before 2023. I doubt that those methods still represent the current SOTA given the rapid advance in the literature of PTQ. More recent benchmark methods (e.g., [1]-[5]) need to be included into the comparison.

[1] Albert Tseng, Jerry Chee, Qingyao Sun, Volodymyr Kuleshov, and Christopher De Sa. Quip#: Even better
llm quantization with hadamard incoherence and lattice codebooks. arXiv preprint arXiv:2402.04396,
2024.
[2] Haoxuan Wang, Yuzhang Shang, Zhihang Yuan, Junyi Wu, and Yan Yan. Quest: Low-bit diffusion model
quantization via efficient selective finetuning. arXiv preprint arXiv:2402.03666, 2024.
[3] Xiuying Wei, Yunchen Zhang, Yuhang Li, Xiangguo Zhang, Ruihao Gong, Jinyang Guo, and Xianglong
Liu. Outlier suppression+: Accurate quantization of large language models by equivalent and optimal
shifting and scaling. arXiv preprint arXiv:2304.09145, 2023.
[4] Junyi Wu, Haoxuan Wang, Yuzhang Shang, Mubarak Shah, and Yan Yan. Ptq4dit: Post-training quantization
for diffusion transformers. arXiv preprint arXiv:2405.16005, 2024.
[5] Guangxuan Xiao, Ji Lin, Mickael Seznec, Hao Wu, Julien Demouth, and Song Han. Smoothquant:
Accurate and efficient post-training quantization for large language models. In International Conference
on Machine Learning, pages 38087–38099. PMLR, 2023.

**Questions:**

1) Did the authors forget to highlight the best-performing results in Table 3? Because they did use bold-type to mark the best results in Tables 4 and 5. Just curious about this apparent inconsistency.
2) What is the model difference between the two settings (Lines 459 and 460)? Since the results show some notable difference, the two models must be different, I think.
3) Any failure cases or the limitations of AdaQTransform? I think an improved understanding needs to cover both positive and negative findings.
4) I am curious about the relationship between this work and Outlier Suppression+ (OS+) framework (EMNLP'2023). Both papers deal with scaling and translating. OS+ also developed a fast and stable scheme to calculate effective shifting and scaling values. It will be nice if the authors can compare their strategy for determining scaling/translation parameters.

---

### Official Review · Reviewer_b342 · 2024-10-29

**Soundness:** 3
**Presentation:** 3
**Contribution:** 2
**Rating:** 6
**Confidence:** 3

**Summary:**

In this paper, the authors propose an Adaptive Quantization Transformation (AdaQTransform) for post-training quantization (PTQ) reconstruction, which provides adaptive per-channel transformations on the quantized output features. They explore the quantization step for weights across six cases by decoupling and incorporating learnable parameters in PTQ, such as Batch Normalization (BN) and bias. Extensive experiments on both low-level and high-level tasks demonstrate the superiority of the proposed method.

**Strengths:**

1. The proposed method technically sounds and produces superior performance.

**Weaknesses:**

1. Figure 1 should be revised, as the presentation of decoupled PTQ is not sufficiently clear. The authors should also highlight the advantages of the decoupled design in the introduction.
2. The RepQ-ViT model reparameterizes scales for hardware-friendly layer-wise quantization and log2 quantization during inference. In this paper, the authors introduce additional per-channel factors, which may not be hardware-friendly.
3. Channel-wise scale factor can enhance the performance of PTQ, particularly in low-bit settings. Can the authors provide performance metrics for 2-bit PTQ on the ViT for classification tasks?
4. For both ViT and CNN architectures, can optimizing scale factors in Batch Normalization and Layer Normalization during PTQ yield similar performance?
5.  It would be beneficial for the authors to include an evaluation of inference speed on the hardware.
6. The authors should cite recent works in the related literature, especially concerning Quantization-Aware Training (QAT), such as: "Wang, Longguang, et al. 'Learnable Lookup Table for Neural Network Quantization.' Proceedings of the IEEE/CVF Conference on Computer Vision and Pattern Recognition, 2022."

**Questions:**

See Weaknesses.

---

### Official Review · Reviewer_YY8F · 2024-11-04

**Soundness:** 2
**Presentation:** 1
**Contribution:** 1
**Rating:** 3
**Confidence:** 4

**Summary:**

This paper presents a network quantization method for post-training quantization (PTQ). The proposed method incorporates learning per-channel scaling and translating by decoupling transformation before and after quantization. Experimental results present that the proposed method outperforms the compared methods in image classification and super-resolution.

**Strengths:**

The proposed method outperforms the compared methods in various backbone networks and tasks (image classification, super-resolution, and large language models).

**Weaknesses:**

The presentation of this paper is very confusing and difficult to follow.
For instance, Figure 1 does not describe “case1.1.1” and s_w’ which make it hard to understand at the beginning of this paper.

The proposed method of per-channel transformation is not clearly described and analyzed.
For instance, the manuscript does not define h(\alpha) in Equation (2). Note that the manuscript should describe the exact operation of AdaRound used in the proposed method. Moreover, the experiments have not presented performance improvement through learning per-channel transformation after quantization and ablation studies of each proposed technique component.

The performance comparisons for image super-resolution in Table 6 incorporate wrong numbers.
EDSR of scale 4 in FP32 can not perform 37.99 dB in Set 5, and AdaBM outperforms the reported scores with lower FAB in the paper.

**Questions:**

Why has the parameter s_w been fixed during the other quantization parameter learning?
Is the parameter b not updated during PTQ?
Is the parameter scale in Equation (13) learnable?

---

### Official Review · Reviewer_A9Zs · 2024-11-04

**Soundness:** 3
**Presentation:** 3
**Contribution:** 2
**Rating:** 6
**Confidence:** 4

**Summary:**

This paper proposes AdaQTransform, a new PTQ approach which provides adaptive per-channel quantization on the output feature and can reduce the PTQ feature reconstruction error. This method achieves a new PTQ SOTA on many network structures like CNNs, ViTs, LLMs, and image super-resolution networks. Experiments with TVM on hardware also demonstrate that this method does not introduce additional inference cost.

**Strengths:**

1. The paper is well-written and easy to follow, presenting a new PTQ method that can be integrated with existing methods into a general quantization framework.

2. Comprehensive experiments across various tasks, models, and datasets demonstrate the method's effectiveness.

**Weaknesses:**

1. This paper does not provide an analysis of the additional training overhead caused by the training learnable parameters $\xi$ and $\eta$ compared to other methods.

2. The hardware experiment only presents the results of the NWQ method on the Mobile-V2 model, which is not sufficient for the practical applications of relevant researchers and practitioners.

**Questions:**

1. This paper only provides the results under weight-activation quantization. How does the AdaQTransform method perform under the weight-only quantization of CNNs, ViTs, LLMs, and image super-resolution networks?

2. What is the training cost of the additional parameters $\xi$ and $\eta$? How does the required number of iterations of the optimization process (to get the parameters $s_x$, $\alpha$, $\xi$ and $\eta$) compared to the existing methods like BRECQ?

3. Can the authors provide more models and quantization methods to compare the inference performance under the same hardware on TVM?

---

### Meta-Review · Area_Chair_cmVP · 2024-12-20

**Metareview:**

Summary:  Proposes a post-training-quantization (PTQ). The main contribution is on unifying the PTQ with QAT with a learnable scale and translation per-channel factors.

Strength: Unifying PTQ and QAT. The paper is overall well written. The method, while not easy to follow for everyone, describes the method in a way that domain experts can follow. The experiments are extensive and strong.

Weakness: There were a number of typos which were corrected after the rebuttal. There were missing comparisons on additional training overhead of the per-channel scale and translation parameters, and inference speed which were addressed in the rebuttal. I encourage authors to revise the paper to include all the newly generated additional ablations and experimental results into the main paper.

**Additional Comments On Reviewer Discussion:**

The paper received 3x marginally above acceptance threshold and 1x reject. Except for reviewer (b342), who maintained a reject rating, two other reviewers’ are convinced by the rebuttal. Although one reviewer did not respond to the rebuttal. I have read the rebuttal which adequately addresses the questions raised. Reviewer (b342) initial questions were also fully addressed as acknowledged by b342. Subsequently, b342 raised additional questions on (1) proper positioning of the work on PTQ vs QAT and (2) on the advantages and disadvantages of the new parameters introduced. The response has convinced me that the introduction of QAT in the methodology improves the understanding of the technique, as also championed by reviewer (8Mvu). For these reasons, I am overwriting one reviewer's concern and recommend acceptance of the paper. See merits of the paper in the strength section too.

---

### Decision · Program_Chairs · 2025-01-22

Accept (Poster)